# Transcriptomic and Co-Expression Network Profiling of Shoot Apical Meristem Reveal Contrasting Response to Nitrogen Rate between *Indica* and *Japonica* Rice Subspecies

**DOI:** 10.3390/ijms20235922

**Published:** 2019-11-25

**Authors:** Xiaoxiang Zhang, Juan Zhou, Niansheng Huang, Lanjing Mo, Minjia Lv, Yingbo Gao, Chen Chen, Shuangyi Yin, Jing Ju, Guichun Dong, Yong Zhou, Zefeng Yang, Aihong Li, Yulong Wang, Jianye Huang, Youli Yao

**Affiliations:** 1Jiangsu Key Laboratory of Crop Genetics and Physiology/Co-Innovation Center for Modern Production Technology of Grain Crops, Yangzhou University, Yangzhou 225009, China; zhangxiaoxiangyzu@126.com (X.Z.); juanzhou@yzu.edu.cn (J.Z.); m2013325@163.com (L.M.); lvminjia@126.com (M.L.); yingbogao_yzu@163.com (Y.G.); shuangyiyin@126.com (S.Y.); gcdong@yzu.edu.cn (G.D.); zhouyong@yzu.edu.cn (Y.Z.); zfyang@yzu.edu.cn (Z.Y.); ylwang@yzu.edu.cn (Y.W.); 2Lixiahe Agricultural Research Institute of Jiangsu Province, Yangzhou 225007, China; jsyzhns@163.com (N.H.); yzlah@126.com (A.L.); 3Zhenjiang Agricultural Research Institute of Jiangsu Province, Jurong 212400, China; Chen852235436@163.com; 4College of Environmental Science and Engineering, Yangzhou University, Yangzhou 225000, China; jujing@yzu.edu.cn

**Keywords:** shoot apical meristem, transcriptomic analysis, co-expression network, tiller, nitrogen rate, rice (*Oryza sativa* L.)

## Abstract

Reducing nitrogen (N) input is a key measure to achieve a sustainable rice production in China, especially in Jiangsu Province. Tiller is the basis for achieving panicle number that plays as a major factor in the yield determination. In actual production, excessive N is often applied in order to produce enough tillers in the early stages. Understanding how N regulates tillering in rice plants is critical to generate an integrative management to reduce N use and reaching tiller number target. Aiming at this objective, we utilized RNA sequencing and weighted gene co-expression network analysis (WGCNA) to compare the transcriptomes surrounding the shoot apical meristem of *indica* (Yangdao6, YD6) and *japonica* (Nipponbare, NPB) rice subspecies. Our results showed that N rate influenced tiller number in a different pattern between the two varieties, with NPB being more sensitive to N enrichment, and YD6 being more tolerant to high N rate. Tiller number was positively related to N content in leaf, culm and root tissue, but negatively related to the soluble carbohydrate content, regardless of variety. Transcriptomic comparisons revealed that for YD6 when N rate enrichment from low (LN) to medium (MN), it caused 115 DEGs (LN vs. MN), from MN to high level (HN) triggered 162 DEGs (MN vs. HN), but direct comparison of low with high N rate showed a 511 DEGs (LN vs. HN). These numbers of DEG in NPB were 87 (LN vs. MN), 40 (MN vs. HN), and 148 (LN vs. HN). These differences indicate that continual N enrichment led to a bumpy change at the transcription level. For the reported sixty-five genes which affect tillering, thirty-six showed decent expression in SAM at tiller starting phase, among them only nineteen being significantly influenced by N level, and two genes showed significant interaction between N rate and variety. Gene ontology analysis revealed that the majority of the common DEGs are involved in general stress responses, stimulus responses, and hormonal signaling process. WGCNA network identified twenty-two co-expressing gene modules and ten candidate hubgenes for each module. Several genes associated with tillering and N rate fall on the related modules. These indicate that there are more genes participating in tillering regulation in response to N enrichment.

## 1. Introduction

As one of the three most important cereal crops cultivated for thousands of years, rice (*Oryza sativa* L.) provides staple foods for nearly half of the world’s population. Keeping its production apace with the increasing demand is critical to food security. The most gain in rice yield is attributable to increased N fertilizer addition to the paddy field. N is an essential macronutrient for completing rice plant growth and development, therefore, usually a limit to its production [1]. Inarguably, N is the most effective fertilizer in promoting crop growth and increasing crop yield [1]. However, the magic effects of N fertilization deceptively lead to excessive application, which gravely contributes to more direct ammonium gas loss, N run-off, water eutrophication, nitrous oxide greenhouse gas emissions and soil acidification [1,2,3]. Consequentially, excessive N fertilization has become a fundamental environmental issue and a health problem. Improving N use efficiency is believed to be the ultimate solution to mitigate these problems. Ideal N application rate and proper timing are the keys to reaching a balance of yield gain and N use efficiency [2,3]. *Indica* and *japonica* rice are two major subspecies in Asia, and reportedly being different in response to N enrichment [4,5].

Rice tiller begins at the leaf sheath auxiliary of the bottom nodes of a host culm, close to shoot apical meristem (SAM) at the early growth stage. Usually, rice tiller first appears when the 4th true leaf emerges. Tiller number at 8-9th week after germination essentially represents a variety’s tiller production feature [6]. Tillers provide more opportunities to the development of more panicles. Among the agronomic yield traits of rice, panicle number is the one that being determined at the earliest stage. More importantly, panicle number largely associates with the spikelet number and filled grain number per panicle. Therefore, reaching a suitable number of tillers timely is a benchmark in setting the cornerstone for achieving an ideal grain yield in rice production.

Meanwhile, as a monocot fully sequenced species, rice provides a great model to decode the molecular secrets of tiller regulation. Recent progress in rice molecular genetics reveals that more than 65 genes at different stages engage in tiller number regulation [7,8,9,10,11,12,13,14]. However, how these genes concert in tiller control remains to be an enigma. High throughput RNA sequencing (RNA-Seq) lends us an effective tool to discover the transcriptomic profile of every active gene [15]. Though much transcriptome data exists in rice, none targets at the tissues near the SAM region, especially in response to N enrichment.

In the present study, we validated the differential tillering responses to N rates of *indica* (Yangdao6, YD6) and *japonica* (Nipponbare, NPB) rice subspecies, performed RNA sequencing and weighted gene co-expression network analysis (WGCNA) to compare their transcriptomes in the surrounding tissues of SAM. The objectives of this study is to compare their differential transcriptomic responses to N rate, and to reveal the concerted molecular network in tiller regulation in rice. The results would expand our understanding of how complicated an important agronomic trait is controlled.

## 2. Results

### 2.1. N Enrichment Promotes Tillers in a Different Pattern between NPB and YD6

Rice tiller depends much on N availability, and hence a substantial portion of N fertilizer is applied before seeding or transplanting in field production. In this study, we enriched N supply at two rates, 9 (MN) and 18 (HN) g N m^−2^, which were comparable to the average and high range of top-dressing level in the local field production practice at this growth stage. Results showed that enriching N rate to MN level promoted tillering significantly at the 4th leaf emerging stage in both NPB and YD6 (Figure 1). However, further increasing N input to HN could not further enhance tillering at this stage. On the contrary, HN even suppressed tillering at the 4th leaf stage, compared with MN. At the 6th leaf emerging stage, MN, and HN treatments produced the same number of tillers in NPB, both significantly higher than that of LN (CK); but the tiller number was not significantly different among LN, MN and HN at the 8th leaf stage in NPB. The tiller number in YD6 was consistently HN > MN > LN at the 6th and 8th leaf stage. These indicate that for producing more tillers at an early stage, varieties like NPB did not require much N enrichment if any, whereas varieties like YD6 required a higher N rate, albeit preferably in a mild rate. Excessive N enrichment did not promote tiller as wished, irrespective of varieties.

### 2.2. Tiller Number Is Related with N and Carbohydrate Content

Analysis of the N and carbohydrate content of the tissues revealed that MN enrichment increased N content in all tissues (root, sheath, and leaf) at the 4th leaf emergence stage, irrespective of varieties (Figure 2). However, further enrichment of N rate from MN to HN did not alter the N content at this stage. At the 6th leaf stage, the N content showed consistently to be HN > MN > LN (CK), irrespective of tissues or varieties. Yet, the gap of N content diminished at the 8th leaf emerging stage. These indicate that MN could enhance N absorption at the 4th leaf emergence stage, but more N enrichment could not promote N absorption further. The differential gap in N content disappeared as the plant grew for two more leaf-age. It is worthy to notice that the N content dropped faster in YD6 than in NPB from 4th to 6th leaf stage, corroborating that YD6 requires more N input to maintain a similar N content than NPB. This suggests different N management strategy should be applied to different variety for keep N content for tillering.

Water soluble carbohydrate (WSC) (i.e., soluble sugar) content was almost comparable between root and sheath, both being much lower than that in leaf (Figure 3A,C). MN enrichment treatment lowered WSC drastically, whereas HN caused slightly further decrease. Similarly, starch content reduced as N rate increased. Yet, the highest starch content was consistently observed in the sheath, with root being higher than that in leaf (Figure 3B,D), irrespective of growth stages. The correlation between N content and starch, soluble sugar are not significant, irrespective of varieties (Figure 3E,F).

The tiller number at the 4th leaf stage was significantly correlated with the tiller number at the 6th leaf stage, but not significantly correlated with those at the 8th leaf emerging stage, respectively (Appendix A), suggesting the association was diluted along with the growth. The tiller number at the 6th leaf stage was positively correlated with the N content in the root (*R^2^* = 0.6303 ^ns^), sheath (*R^2^* = 0.8783 **) and leaf (*R^2^* = 0.0.8719 **) at this stage (Appendix A). It displayed a negative correlation with WSC in the sheath (*R^2^* = 0.3970 ^ns^) and leaf (*R^2^* = 0.7956 **) (Appendix A), and also negative correlation with the starch content in the root (*R^2^* = 0.6677 *), sheath (*R^2^* = 0.4848 ^ns^) and leaf (*R^2^* = 0.7036 *) (Appendix A). These generally confirm that the tiller number correlated positively with the N content, but negatively with WSC and starch content.

### 2.3. RNA-Seq Data Quality and Assembly

To reveal the transcriptomic change responsive to N rates, cDNA libraries from tissues surrounding SAM were constructed and subjected to RNA-Seq analysis. A total of 624 million raw reads were generated from two subspecies *japonica* NPB and *indica* YD6 in combination with three levels of N rate (Appendix A). After removal of adaptor sequences, short reads, low quality, and rice ribosome RNA reads, clean reads ratio proved to range from 91.6% to 97.1%, indicating that the sequencing reads were qualified for further analysis.

These clean reads were mapped to the NPB reference genome, the mapped ratio ranged from 97.3% to 98.1% in NPB samples, and from 90.9% to 93.5% in YD6. Since the most recent physical map covers about 97% of the NPB genome [16], our mapped ratio of the transcriptomes in SAM tissues in NPB was generally consistent with it. These also indicate that the alignments were successful. As expected, the transcriptome from YD6 was genetically more heterogeneous to NPB reference genome. Even when trying to map the clean reads of YD6 to a reference genome of 9311, a sister line of YD6, the successfully mapped ratio for YD6 ranged from 93.5% to 95.9%. For comparison consistency, we used the physical genomic map of NPB as a common reference genome in all of the following analysis.

Principal component analysis (PCA) projects the whole transcriptome profile onto a few variables to reflect the distribution across samples, which are often clustered in two or three-dimensional space [17]. PCA revealed that transcriptome profile between YD6 and NPB were more distant than that among N rates (Appendix A), reflecting that the variety defined the most range of the differences. The variability of LN and MN were less than that of HN in NPB, while MN was less variable than that of LN and HN in YD6. Yet, the general variability in NPB was much less than that of YD6. These suggest that the transcriptomic response to N rate was more variable in YD6 than in NPB, the transcriptomic variability caused by N enrichment also depends on the plant’s existing N rate condition, and the transcriptomic differences between varieties were more distant than their responses to N rate.

The correlation analysis of gene expression between the biological replicates reflects the experimental and sequencing reproducibility, whereas the correlation between the groups reflects the variety and N rate effects. The correlation coefficients were determined by the FPKM values of the individual genes to verify the general reproducibility and variability among replicates and groups, respectively (Appendix A). The correlation coefficient *R* value between the biological replicates ranged from 0.96–0.97 in NPB and 0.95-0.96 in YD6, respectively, exhibiting a remarkably high consistency. This meant this set of RNA-Seq results had very high reproducibility in replicates. Meanwhile, the *R* value between NPB and YD6 at corresponding N rate ranged from 0.89–0.92, displaying certain consistency. Yet, these values were lower than the *R* between the biological replicates, indicating more variability between varieties, as expected. Furthermore, the heat map between the groups is consistent (Appendix A). These ensure that the RNA-Seq generated a highly reproducible expression data among the replicates and a proper variability across the comparison groups, which corroborated with PCA results as well as phenotypic performance.

### 2.4. Validation of Selected DEG Confirms RNA-Seq Data Reliability

Twenty-one DEGs of different FPKM levels were selected for validation by qRT-PCR (Appendix A). The quantification data were used to correlate with the FPKM value of each treatment. Among them, twenty genes showed a positive linear correlation between the FPKM and qRT-PCR data, with the correlation coefficient *R^2^* ranging from 0.6785 * to 0.9721 ** (Figure 4). One gene could not be amplified due to the primers failure. This affirmative validation result indicates that the transcriptome profiling generally reflected the virtual transcript expression differences in the experiment.

### 2.5. Differentially Expressed Genes (DEGs) in Response to N Rate

To identify the DEG in response to N rate, Cuffdiff [18] program was used in comparing the FPKM of the treatment vs. control according to the following criteria: fold change (FC) |log2| ≥ 1.0 and false discovery rate (FDR, *q*-value) ≤ 0.05. The total DEG in response to N enrichment in NPB were 87 (65 down-regulated and 22 up-regulated), 40 (12 down and 28 up) and 148 (69 down and 79 up) between LN and MN, MN and HN, LN and HN comparisons, respectively (Figure 5A,D). Among them, merely four genes were consistently present in every comparison. The corresponding numbers of DEG were 115 (58 down and 57 up), 162 (80 down and 82 up) and 511 (239 down and 272 up) in YD6, and only 15 genes were commonly present in these comparisons (Figure 5B,E). The number of DEG indicate that more drastic transcriptome changes from LN to HN, rather than from LN to MN, or MN to HN, irrespective of varieties. 

Apparently, the number of DEG in response to N enrichment was much more in the *indica* variety YD6 than in the *japonica* variety NPB. Surprisingly, of those DEG commonly presented in response to N rate in NPB and YD6, only one common gene was found (LOC_Os01g01660), which putatively encodes an isoflavone reductase-like gene and confers tolerance to reactive oxygen species. This scarce overlapping in the DEGs suggested that the transcriptome responses to N enrichment were very much dependent on the different genetic background. Combination of different sets of background genes may totally alter the transcription profile in response to certain N enrichment.

On the contrary, comparison between NPB and YD6 at each respective N rate revealed that the number of DEG commonly presented at each N level was a majority of 463 genes (Figure 5C,F), much more than those numbers of distinctive DEGs at each N rate. Among these DEGs, we listed several specific types of genes (Figure 5F–L). This corroborates that the varietal set of DEG was a core determinant in shaping their response to various N enrichment.

### 2.6. Gene Ontology (GO) Analysis and KEGG Clustering

Web-based String (version 10.5) program was employed to define the classification of N-responsive DEG involved in the three domains of the biological process, cellular component and molecular functions, and their KEGG pathway clustering (Appendix A). In the comparison of LN vs. HN in NPB, the number of DEG in the category of biological process was in the order of the cellular process (41), metabolic process (34), single-organism process (32), response to stimulus (30) and biological regulation (22). Of the DEG fell into the category of cellular component, the order was the cell (44), cell part (44), organelle (35) and membrane (14). Of the DEG that fell into the category of molecular function, the order was the binding activity (23), catalytic activity (16), transporter activity (four), and transcription factor (TF) activity (four, Appendix A). Similar orders in the number of DEG were also revealed in LN vs. HN comparison in YD6 as well as in other N rate comparisons, and the comparisons between varieties at the respective N rate (Appendix A, Appendix A). Based on the number of DEGs, more genes are concentrated in the biological process, followed by cellular components, and finally in molecular function. Apparently, these biological procedures were the major changes in the tissues near SAM in response to N rate.

KEGG clustering of the DEG revealed that the first seven categories of common pathways between NPB and YD6 responsive to N rate were the metabolic pathway, ribosome, plant-pathogen interaction, microbial metabolism in diverse environments, plant hormone signal transduction, antenna proteins in photosynthesis and protein processing in the endoplasmic reticulum (Appendix A). The major differential pathways between NPB and YD6 were the glutathione metabolism, protein export, and selenocompound metabolism, which were unique to NPB, and 13 pathways unique to YD6 including the biosynthesis of secondary metabolites, photosynthesis, phenylpropanoid biosynthesis and phenylalanine metabolism. These indicate that the common basic pathways (metabolic and ribosome) were important for both varieties’ response to N rate, and many unique pathways were involved in the procedures as well.

### 2.7. Alternative Splicing Transcripts and Novel Genes

Alternative splicing (AS) creates more versatile regulating RNAs and translatable mRNAs, thus a more enriched diversity in transcriptome and possible code proteins [19]. In this experiment, a total of 68,385 AS events, derived from 12,044 annotated genes, had been revealed (Figure 6A,B). Among them, the retained intron was the most common event type (34.4%–34.9%), followed by the alternative 3′ splicing sites (31.3%–31.7%), alternative 5′ splicing sites (17.6%–17.8%), skipped exon (12.7%–12.8%), a mix of two or more different types of AS (complex, 2.6%) and mutually exclusive exons (0.8%). The number of AS events between the varieties and among N rates did not show any statistically significant difference. However, the AS events relative to clean reads in all category type was consistently 1–4% higher in NPB than that in YD6, which probably reflected the varietal differences.

Totally 352 novel genes were found in the RNA-seq data. Among them, 74 and 174 genes were unique to NPB and YD6, respectively; 104 genes were common in the two varieties (Figure 6C). Novel genes responsive to LN were 58 and 89 specific to NPB and YD6 respectively; 55 were common in the two varieties (Figure 6D). These corresponding novel transcripts numbers were 48, 102 and 60 in MN, 48, 93 and 52 in HN, respectively (Figure 6E,F). These results suggest that deep sequencing may reveal more tissue-specific transcripts. Comparison of the novel gene number among N rates within a variety also revealed similar trends, though the numbers were always bigger in YD6 than that in NPB (Figure 6G,H). The novel genes revealed were always more in the *indica* variety YD6 than the *japonica* variety NPB, probably partially attributable to that the transcript annotation in the *japonica* subspecies was more fully covered in the reference genome.

### 2.8. Identifition of Weighted Gene Co-Expression Network

To obtain candidate key genes associated with the phenotypical traits, WGCNA [20] was used to distinguish the specific genes that are related to the traits, including tiller number, N content in leaf and in stem, dry weight of root, stem and leaf etc. After removing the genes with low FPKM levels, 21,700 genes were retained for the further WGCNAs analysis. This analysis identified twenty-two distinct gene co-expression modules (labeled in different colors) shown in the dendrogram (Figure 7A), and a total of eleven modules (brown, green, darkred, red, blue, magenta, pink, tan, lightyellow, turquoise and purple) were significantly related to the above-mentioned various phenotypical traits, respectively (Figure 7B). The eigengene expression in blue and green modules (M10 and M3) are shown in Figure 7C,D, respectively. The tiller number was positively most correlated with the eigengene expression levels of the lightyellow module (M18), and the correlation coefficient (*r*) was 0.61 (*p* = 0.04). Gene ontology (GO) annotation analysis reveals that the cellular components of this module genes mostly fall into extracellular region, yet molecular functions and biological processes are not significantly enriched (Appendix A). The tiller number was negatively most correlated with the green module, with *r* = −0.71 (*p* = 0.01). GO annotation shows that the biological processes of the green module genes fall into biological regulation, regulation of cellular process, transport, establishment of localization, localization, regulation of transcription, etc. The molecular function fall into transporter activity, protein binding, transmembrane transporter activity, active transmembrane transporter activity, and transcription regulator activity. The cellular component of this module fall into membrane, integral to membrane, intrinsic to membrane, and cell part (Appendix A).

The scales of N content in stem and in leaf were highly positively correlated with the eigengene expression in the blue module (M10), with *r* at 0.71 (*p* = 0.01) and 0.69 (*p* = 0.01), respectively. GO annotation shows that the biological processes of the blue module genes fall into the translation, gene expression, cellular protein metabolic process, small molecule metabolic process, protein folding, protein metabolic process, cellular ketone metabolic process, carboxylic acid metabolic process etc. The molecular functions of the blue module are the structural constituent of ribosome, structural molecule activity, translation factor activity, nucleic acid binding, pyrophosphatase activity, and GTP binding. Their cellular components are cytoplasm, macromolecular and ribonucleoprotein complex, ribosome, and cytoplasmic part (Appendix A). And the scale of N content in stem and in leaf were all negatively correlated with the green module (*r* = −0.81, *p* = 0.002) and (*r* = −0.74, *p* = 0.006), consistent with the negative correlation module of the scale of tillers. In addition, the modules related to other traits and their GO annotations are shown in Figure 7B and Appendix A. Interestingly, modules positively related with N content are in a negatively association with phosphorus content, and vice versa for the positively related modules. Their associations with sulphur, calcium and potassium content are in a similar pattern but not that consistent. These reveal that the phenotypic traits are more closely associated with certain modules of genes than other modules.

### 2.9. Construction of the Gene Co-Expression Networks and Identification of Candidate Hub Genes

After screening for the highest significant correlated modules, we selected two modules (one positive correlation module and one negative module) to construct the gene expression networks (Figure 8, Appendix A). Since the scale of N in stem and N in leaf, dry weight in stem and in leaf share the same selected modules, totally seven most relevant modules (light yellow, green, blue, tan, red, brown and pink module) were chosen for the further analysis. The association degree of eigengenes within the module was used to construct the gene expression network (Figure 8A,B, Appendix A). The hub genes are selected by K_ME_ values through their closest connections in the gene network. The top 20 node genes of each module were screened to generate a network map (Figure 8C,D, Appendix A).

To screen for candidate hub genes, the top ten connected genes in the specific modules were picked out by their K_ME_ value (Table 1, Appendix A). The tiller number relate positively with the lightyellow module (M18) and negatively with the green module (M3), respectively (Table 1). In the light yellow module, the top ten genes encode HEV3—Hevein family protein precursor, BBTI5, BBTI8, VP15, ribonuclease T2 family, pyridoxal-dependent decarboxylase protein, cysteine-rich receptor-like protein kinase, and kinesin motor domain containing protein. The correlation network of the light yellow module is shown in Figure 9A. Genes encoding the aforementioned proteins were identified as key candidate hub genes for the light yellow module (M18) (Figure 8C). In the green module (M3), the top ten genes encode MATE efflux protein, ethylene-responsive protein, protein kinase, GDSL-like lipase/acylhydrolase, aspartic proteinase oryzasin-1 precursor, eukaryotic translation initiation factor 1A, thaumatin, tetraspanin family protein, AGAP002737-PA, and RFC5. Genes encoding these proteins were identified as candidate hub genes for the green module (Figure 8D). Similarly, multiple functional proteins were enriched in this module, indicating that these regulatory networks may play a pivotal role in regulation of the scale of tillers.

The N content in stem and in leaf relate positively with the blue module and negatively to the green module, respectively (Table 1). In the blue module, the top ten genes encode RNA recognition motif containing protein, ribosomal L18p/L5e family protein, cytochrome P450, ribonuclease T2 family, heat shock 22 kDa protein, CCT motif family protein, NB-ARC domain containing protein, and transcription factor etc. The correlation network of the blue module is shown in Appendix A. This indicates that these genes may play a crucial role in regulation of the scale of N in stem and in leaf. Interestingly, the negative module falls into the same module with the scale of tillers. In addition, the co-expression network and candidate hub genes in other modules are shown in Appendix A and Appendix A.

It is noteworthy that we find four hub genes overlap with DEGs (between varieties) in the turquoise module (Appendix A), which negatively correlated with the scale of dry weight in leaf. These four genes are LOC_Os12g07830, LOC_Os06g15570, LOC_Os08g25050 and LOC_Os06g12170. The results suggest that these genes in the regulatory network may play a core role in differential dry weight accumulation between varieties.

The heatmap and changes in the expression levels of the candidate hub genes from the selected modules are shown in Figure 9 and Appendix A. The expression level of these hub genes are contrastingly different between the varieties, indicating that the hub genes may play a crucial role in their differential response to N rate between the varieties.

### 2.10. Expression Profiles of Tiller Related Genes and Their Network

Many genes are previously reported to impact upon tiller numbers in their mutants [9]. To reveal their transcriptional responses to N rate, we extracted their FPKM value from the transcriptomes, and subjected it to statistical comparison (Table 2). Totally of 65 genes were selected to screen for their transcription levels. Among the 36 decently expressed genes (FPKM > 0.1), 27 showed significant varietal difference (*p* ≤ 0.05), 18 showed to be responsive to N rate, and three showed significant interaction between the variety and N rate. Of these significantly influenced tiller genes, only six of the 27 genes in the varietal comparison and nine of 18 genes in the N rate comparisons displayed more than twofold change in at least one comparison. Apparently, the expression level of these genes was more determined by a variety than to be affected by the N rate. A protein and protein interaction network (PPI) constructed from known connections seems to illustrate their complexity more (Figure 10). However, the change did not need to surpass a threshold of twofold to produce a significant impact on the phenotypes such as the tiller number.

Meanwhile, through co-expression network analysis, these tiller related genes belong to different modules, and *D10/OsCCD8*, *HTD2/D14/D88/qPPB3*, and *NRR/CRCT* genes fall in the green module, which negatively controls the occurrence of the tiller number (Table 2). Surprisingly, these tiller genes are not part of the candidate hub genes. This suggests that these tiller related genes are not at the nodes of the gene expression network, which may locate downstream of certain pathways, and changes of hub genes will regulate their expression accordingly. In addition, we found that other tiller-related genes belong to different network regulation modules, showing the complexity and diversity of the gene regulation at the early stage of tillering. It seems that many genes have other functions that are not yet discovered during this growth stage. For example, *TAC, LAX* and *IPA1* fell in the turquoise module are negatively correlated with dry weight in the leaf during this period.

### 2.11. Expression Profiles of No Apical Meristem Family Genes, Carbohydrate, and N Metabolism and Transport-Related Genes

No apical meristem family genes (NAM) are critical for meristem maintenance and floral development [21]. Tillers contain newly branched meristems and therein arise perspective panicles. Rice genome contains 90 putative NAM genes. Among the 76 NAM genes detected in the transcriptome at this stage, 27 had an extremely low expression (FPKM ≤ 0.1). Among the 49 decently expressed NAM genes (FPKM > 0.1), 18 displayed significant varietal differences, 12 being responsive to the N rate, and 10 showed an interaction between the variety and N rate (Appendix A). Among those significantly changed, only 11 of the 18 genes in the varietal comparison and six of the 12 genes responsive to the N rate displayed a twofold or higher change at least once. Apparently, more NAM genes were defined by the variety than to be responsive to the N rate.

Carbohydrates provide basic carbon substrates for multiple primary and secondary metabolic pathways, and are subject to the impact of N availability [22,23]. Among the totally 555 carbohydrate metabolism-related genes we screened for in the transcriptome data, 29 were not detected in the SAM at this stage, and 45 showed an extremely low expression (FPKM < 0.1). Of those 482 decently expressed genes, 200 genes did not show significant (*p* > 0.05) differences in any comparison; 212 showed significant varietal differences (*p ≤ 0.05*, Appendix A); 128 being responsive to the N rates; and 43 showed significant interaction between the variety and N rate. This corroborates with that more genes were expressing differentially between the varieties than those in response to the N rate. All these specific pathway profile analysis points toward that the expression profile differences were more likely to be determined by a variety rather than the N enrichment.

N enrichment directly affects the N availability, absorption, synthesis and transport of the related amino acids and the downstream N metabolism [2,24]. We screened for 210 genes pertaining to ammonium, nitrate, nitrite, glutamate, glutamine, asparagine, tryptophan, methionine, aspartate, proline, glycine, cysteine and NADH metabolism. We detected 199 genes expressed in the SAM, with 178 showed a decent expression (FPKM > 0.1). Among them, 97 showed significant differences between the varieties; 69 showed responsive to the N rate; 28 displayed significant interaction between the variety and N rate (Appendix A). Comparably, much more genes expressed differentially between the varieties than those being responsive to the N rate. At the same time, we found that the genes related to N metabolism are in different modules and play different roles during this period (Figure 10). The N transporter proteins, *OsNRT1* and *OsNRT2.3/OsNRT2.3a/OsNRT2.3b*, are in the brown module which positively relate to dry weight of stem and leaf (Table 3). Interestingly, an ammonium transporter protein-coding gene *OsAMT1.2* locates in the turquoise module, which negatively regulates the dry weight of leaf. These results show that the N related genes in rice plant might play very different roles in the absorption, function and distribution of N in response to N rate as well as in tillering regulation.

## 3. Discussion

SAM differentiates all organs, and initiates new branches/tillers for more panicles. Understanding the molecular mechanism regulating SAM is of great importance to improve N use efficiency in many crop species, especially in rice. There are so many transcriptomic researches in rice plant. However, surprisingly, none is dealing with this specific important tissue. NPB and 9311 (sister line of YD6) are the two representative *japonica* and *indica* rice varieties being deeply sequenced and several mutant collections are derived from them. In this study, we utilized the transcriptome and co-expression network to analyze the different response to N rate between *indica* and *japonica* rice (YD6 and NPB) at the early stage of tillering occurrence. This study also reveals that the hub genes, tiller genes, NAM genes, and N related genes in each module are playing specific roles.

### 3.1. Reducing N Input to Low or Moderate Rate Is Still Good to Promote Enough Tillers

*Indica* and *japonica* rice subspecies possess much differences in morphological, physiological and cultivation characteristics especially in some important agronomic traits such as tillering. Tiller feature of a specific variety is the product of its genetic background, cultivation practices and environmental conditions. Among the later factors, N availability acts a key role [25,26]. Tiller is the basis for achieving panicle number which plays as a major factor in yield determination. In actual production, excessive N is usually applied aiming to produce enough number of tillers in the early growth stages. Therefore, to reduce N without yield penalty lies much in achieving enough tiller number at a minimal N requirement. Our results suggest that, even for varieties like NPB and YD6 being very different in their sensitivity to N enrichment, they both consistently show that a mild N enrichment can enhance tillering; however, excessive N enrichment will not promote more tillers as wished, instead it even suppresses tillering for a short term effect. Yet, for a more N tolerant variety like YD6, a frequent mild N enrichment may be necessary to boost tillering. Therefore, for reducing N input to a paddy field, cutting N rate to avoid heavy topdressing and switching to a low to moderate N enrichment can serve the same effect on promoting tillers. Meanwhile, at a reduced N rate, it can significantly cut off N run off as well as raise N use efficiency [25]. 

Meanwhile, we found the genes related to N uptake, transport, and N metabolism mainly fall in the blue, brown and turquoise modules through co-expression network in this study (Appendix A). Among them, the brown module correlated positively with the dry weight accumulation. *OsNRT1* encodes a low-affinity nitrate transporter and belongs to the constitutive expression of the outermost layer of roots, epidermis and root hairs [27]. This gene is not only homologous to the *CHL1* (*AtNRT1*) gene in Arabidopsis, but also to the polypeptide transporter that widely present in plants, animals, fungi, bacteria etc. *OsNRT2.3a* plays an important role in the long-distance transport of nitrate from root to shoot in the low-nitrate supply condition [28]. *OsNRT2.3a* is a rice vascular-specific, NRT2 family high-affinity nitrate transporter. *OsNAR2.1* can interact with *OsNRT2.1/2.2* and *OsNRT2.3a* to promote nitrate uptake by rice roots at different nitrate supply levels [29]. *OsNRT2.3b* can enhance the buffering capacity of rice to pH status, increase the absorption of N, iron and phosphorus, improve the effective utilization of N, and is very important for plants to adapt to different forms of N sources [30]. In our study, though *OsNRT1*, *OsNRT2.3* and *OsNPF2.4* showed similar response pattern to N rate, *OsAMT1.2* showed an even different response pattern to N rate between varieties. In previous transcriptome and co-expression network analysis, *NRT* and *AMT* play an important role in N uptake and utilization efficiency in *Brassica juncea* cultivars and N transporters regulate some aspects (shoot or leaf) of the coordination of N and C metabolism in *Arabidopsis* [31,32]. Interestingly, these genes have a close positive regulation of dry weight of stem and leaf in this study, which suggests that these genes may share common roles in regulating N transportation and biomass. These results indicate that these genes have a promoting effect on the growth and development of the above-ground organs. Although it does not belong to hub genes, it plays a role in the transport and absorption of N under the control of upstream hub genes.

### 3.2. Most of the Tiller Genes Are Not Drastically Respond to N Rate

Recent progress in molecular genetics have deciphered that *MOC1* [9], *LAX1* and *LAX2* [12], *OsFC1/OsTB1* [7,8], *APC/C^TAD1/TE^* [11,12], and more than 60 other genes [13,14] are involved in tiller number control in rice. However, it is almost unknown which set of genes are involved in tiller regulation under general cultivation conditions, especially which ones are responsible to N rate. In an attempt to answer such a question, we turn to RNA-seq approach to compare the transcriptomic changes in the tissue around the SAM, where the tillers emerge.

Our RNA-seq data point to that drastic N enrichment caused more dramatic transcription change rather than mild N rate (Appendix A). The top categories of those DEGs fall into the metabolic and hormonal/signal transduction pathways. However, all changes of the tiller related genes did not reach the two-fold bar to be a DEG (Table 2). Furthermore, half of these tiller genes were showing opposite change between the two varieties in response to the N rate (Table 2). These suggest us that drastic N rate may trigger a dramatic global transcription shift, but those changes may not necessarily directly relate to tillering. More importantly, change in a gene expression level does not have to surpass the two-fold bar to impact on tiller number.

In addition, previous studies have shown that the tiller genes are mainly analyzed in a certain genetic background. Moreover, the interaction network or regulatory mechanisms of tiller genes at specific stages are not well understood, especially in the early stages of tillering occurrence around the SAM in rice. In this study, we use GWCNA methods to find co-expressed gene modules and explore the relationship between gene networks and phenotypes, as well as core genes in the network. Interestingly, we found that several tiller genes (*HTD2/D14/D88/qPPB3, D10/OsCCD8 and NRR/CRT*) fall in the green module, which is negatively correlated with tiller regulation (Figure 7B). *HTD2/D14*, which is a component of the stragolactones (SLs) signaling pathway, encodes an esterase that inhibits rice branching and negatively regulates rice tiller number [33]. *D88,* function through the MAX/RMS/D pathway, is expressed in most tissues of rice, including leaves, stems and roots and ultimately affect rice plant type by regulating cell growth and organ development. Mutations in *D88* affected the expression of genes involved in tiller formation, including *HTD1, OSH1, D10* and other genes were significantly up-regulated in *d88* mutants [34]. *D10* is a rice ortholog of MAX4/RMS1/DAD1, which encodes a carotenoid cleavage dioxygenase and is involved in the biosynthesis of levodolactone/levylactone derivative SLs [35]. *D10/OsCCD8* is involved in the synthesis of rice aboveground branching inhibitors, and transcription of *D10* may be a key step in regulating the branching inhibition pathway. The interaction among *D10*, auxin and cytokinin affects lateral bud elongation in rice [36]. *NRR/CCRT* can regulate the structure of rice roots, so that they can better absorb a large number of nutrients and play a negative regulation role in the growth of rice roots. It can respond to the level of photo-contracted compounds and coordinate the expression of genes related to starch synthesis [37,38]. Although *HTD2/D88* and *D10/OsCCD8* genes belong to the negative regulatory module and they are not key nodes of regulation network, these genes are regulated by the upstream hubgenes. Our results corroborate that these genes closely relate to tillering.

Meanwhile, our results showed that some genes related to tillering, N and other genes fall in other modules and these modules are not closely related to certain traits (tiller, N, etc.). We speculate that these genes may have some novel unknown molecular functions, which of course requires further experimental verification.

Obviously, these differences between varieties are the results of common regulation of many genes. Changes of certain pivotal gene expression will lead to changes in other genes. The difference in these expression patterns led to the difference in the response of N rate between *indica* and *japonica* rice.

In addition, consistent with previous reports on AS of pre-mRNA [39,40,41,42,43], the major category of AS events revealed in our current study is intron retention (RI), irrespective of rice varieties. Despite the trending differences in the percentage of AS events between the two varieties, we have not found significant distribution differences among the categories event-wise. However, as we did not compare the components of AS genes between the groups, it would be interesting to further analyze if there are differences in this aspects of AS events of certain genes between varieties or among N rates [41,42,43].

## 4. Materials and Methods 

### 4.1. Plant Materials

Two representative rice varieties Nipponbare (NPB, *Oryza sativa* L. subsp. *japonica*) and Yangdao6 (YD6, *Oryza sativa* L. subsp. *indica*) were employed in this study. NPB and 9311 (a sister line of YD6) are widely being used as reference sequencing varieties, in molecular genetic analysis and practical rice production. NPB generally produces more tillers under lower N rate; whereas YD6, bears fewer tillers, yet produces more tillers under higher N rate.

### 4.2. Growth Conditions, N Rate Treatment and Measurement

The experiments were conducted at the Yangzhou University, Jiangsu Province of China. Plants were grown in plastic pots (29 cm in diameter, 30 cm in height), which were filled with a mixture of soil and vermiculite at 5:1 (*v/v*) ratio. The soil type is sandy loam, containing 1.02 g·kg^−1^ total N, 22.73 mg·kg^−1^ available phosphorus, 49.24 mg·kg^−1^ available potassium, and 13.98 g·kg^−1^ total organic matter. The soil osmotic conductivity was 0.11 ms·cm^−1^ and pH was at 7.66.

Total N fertilizer (urea) enrichment rate was set up at 0 (LN, CK), 9 (MN) and 18 (HN) g N·m^−2^. Basal N fertilizer (50% of total rate, in the form of urea, dissolved into water first then further diluted and applied; the same practice was followed for all other top-dressings) was premixed into the top 10 cm of soil at 3 days prior to seeding. Top dressing was at 2nd and 4th leaf emerging stage, each at 25% of the respective N rate. Twelve sprouting seeds were planted in each pot at an even spacing. After germination, the pot was irrigated with a shallow layer (1–2 cm) of water. Four biological replicates were included, with 16 pots in each treatment.

Enumeration of tiller was conducted at 4th, 6th and 8th leaf emerging stage. Plants were sampled for dry matter measurement, and subsequently for soluble sugar, starch and N content determination. Total soluble sugar and starch detection were by anthrone reagent following Brooks et al. [44]. N determination was by Kjeldhal method with Kjeltec 8400 Analyzer Unit (Foss Analytical AB, Hoganas, Sweden) following the manufacturer’s recommendation.

### 4.3. Samples Preparation for RNA Extraction, cDNA Library Construction and Sequencing

To investigate the transcriptome changes in response to N enrichment, tissues for RNA isolation were collected when 4th leaf started spreading out. After removing leaves and the out layers of sheaths with a surgical blade, only tissues surrounding the shoot apical meristem (SAM, about 5mm in length) were collected. SAM isolation operation was carried out on the ice, and the SAM tissues were wrapped in aluminum and snap frozen in liquid N_2_ before transferring to a deep freezer at −80 °C till use.

A total of 12 tissue samples were collected for RNA-Seq, representing two varieties (NPB and YD6) and three N levels (LN, MN and HN), with two biological replicates in each combination. Another set of 12 samples was collected for validation purpose. Total RNA was extracted using the RNAiso Plus Total RNA extraction reagent (Cat#9109, TAKARA, Kusatsu, Japan). Qualified total RNA was further purified by RNeasy micro kit (Cat#74004, QIAGEN, Duesseldorf, Germany). The rRNA was removed using Ribo-Zero rRNA Removal Kits (CAT# MRZMB126, Epicentre, Illumina, San Diego, CA, USA). RNA integrity was evaluated using the Agilent 2100 Bioanalyzer (Agilent, Santa Clara, CA, USA). Samples with RNA Integrity Number (RIN) ≥ 7 were subjected to the subsequent sequencing reactions. The libraries were constructed by using TruSeq Stranded mRNA LT Sample Prep Kit (CAT#15032612, Illumina, San Diego, CA, USA), and sequenced on the Illumina sequencing platform HiSeq 2500, consequentially 100–150 bp paired-end reads were generated.

### 4.4. RNA-Seq Data Processing and Gene Expression Calculation

Raw reads were first filtered by Fastx program to remove disqualified, short or ribosome RNA reads, resulting in clean reads. Clean reads % was calculated as (clean reads/raw reads) %; mRNA % was as ((clean reads-rRNA reads)/clean reads) %; Subsequent genome mapping was by spliced mapping algorithm in Tophat program [45] to a reference genome, *Oryza sativa japonica* NPB version 7.0 from ftp://ftp.plantbiology.msu.edu, and generated BAM files.

To estimate the abundance of gene expression, reads number of uniquely mapped genes were normalized to Fragments Per Kilobase of exon model per Million mapped reads (FPKM) by Cufflinks program [46]. FPKM was calculated as (total exon fragments in reads)/((mapped reads in millions) × exon length in Kilo base pair)), where total exon fragments was the number of reads mapped to exons, mapped reads was the number of reads mapped to the reference genome, and exon length was total base pair number of exons in Kilo base pair.

To generate a list of differentially expressed genes (DEG) between treatments, FPKM was used to calculate the fold change and false discovery rate (FDR, an adjusted *p*-value, i.e., *q*-value) by Cuffdiff program [18]. The threshold of DEG was set as *q*-value≤ 0.05 and fold change ≥ 2.

### 4.5. Novel Gene, Alternative Splicing and Enrichment Analysis

When using Cuffcompare algorithm from Cufflinks software in mapping, reads that could not be mapped to the reference genome and the transcript FPKM ≥ 10 in any single sample were deemed as novel genes. For alternative splicing (AS) screening, the BAM files were processed by Cufflinks program to generate assembled transcripts followed by Cuffmerge algorithm to the final transcriptome assembly, which was subjected to Astalavista (version 3.1) algorithm to reveal AS [19,47,48]. Functional enrichment analysis of gene ontology (GO) and KEGG (Kyoto Encyclopedia of Genes and Genomes) were by String 10.5 [49].

### 4.6. Quantitative Real Time RT-PCR Validation

To validate the RNA-Seq data, twenty-one DEG at different expression level were selected to confirm their expression in the corresponding samples by quantitative real-time reverse transcription polymerase chain reaction (qRT-PCR) with a BioRad CFX-96 system (BioRad, Hercules, CA, USA), following the method in [50]. Briefly, one μg of total RNA from the same batch of RNA for high throughput RNA-Seq was used for the first strand of cDNA synthesis using iScript (Cat#1708891, BioRad, Hercules, CA, USA) according to the supplier’s protocol. The PCR reaction was conducted in a total volume of 12 μL reaction mix, with one μL of cDNA template, 400 nM forward primer, 400 nM reverse primer and six μL of SsoFast EvaGreen Supermix (Cat. #1725200, Bio-Rad, Hercules, CA, USA), and three technical replicates. Two tubulin genes (LOC_Os11g14220 and LOC_Os03g51600) were deemed as the reference genes. Gene-specific primer sequences for qRT-PCR are listed in Appendix A. Quantification was determined by BioRad CFX manager software (V3.1). Fold change relative to control level was determined by the 2^−ΔΔC*t*^ method [50]. PCR amplifications of each sample were used in triplicate.

### 4.7. Co-Expression Network Analysis for Construction of Modules

WGCNA (v1.29) package in R was performed to construct the gene co-expression regulation network [20]. Detailed analysis procedures and methods were followed in accordance with Zhang et al. [51]. Among the 37,824 genes, 21,700 genes with an averaged FPKM from three replicates > 1 were used for the WGCNA unsigned co-expression network analysis. Through testing the independence and the average connectivity degree of different modules with different power value, the appropriate power value in this study was determined as seven. The modules were obtained by the automatic network construction function with default parameters in the WGCNA software package. The correlation between the modules and traits were calculated by the Pearson method using *blockwiseModules* function. The top ten genes with maximum intra-modular connectivity were considered as “highly connected gene” (hubgene) [52]. The top 20 genes including candidate hub gene network was visualized by the Cytoscape (version 3.7.2) [53].

### 4.8. Statistical Analysis of Genes Expression Data in a Specific Pathway

Multivariate analysis procedure of the general linear model (GLM) method from the IBM SPSS software (version 22, IBM, Armonk, NY, USA) was used in the ANOVA comparison of selected gene expression differences. Correlation analysis was made by the correlate procedure in the SPSS software.

## 5. Conclusions

This is a pioneer study to reveal the transcriptomic changes of SAM tissue in response to N rate between *indica* and *japonica* rice subspecies, especially for cultivars widely used in the production. N rate influenced tiller number in a different pattern between varieties, with NPB being more sensitive to N enrichment, and YD6 being more tolerant to N rate. Tiller number was positively related to N content in leaf, culm and root tissue, but negatively related to the soluble carbohydrate, irrespective of variety. Higher N enrichment brought more drastic transcription change than moderate N rate; however, varietal background dominated the differences. For the reported 65 tiller genes, less than half of them showed decent expression in SAM at tiller starting phase; among them only nineteen being significantly influenced by N rate, and two genes showing significant interaction between the N rate and variety. GO analysis revealed that the majority of these common DEGs are involved in general stress responses, stimulus responses, and hormonal signaling process. WGCNA network identified specific modules that are associated with the phenotypic traits and candidate hub genes for each module. Several genes associated with tillering and N content fall on certain most relevant modules. These results help us understand the complexity regulatory mechanisms involved in *indica* and *japonica* rice response to N rate.

## Figures and Tables

**Figure 1 ijms-20-05922-f001:**
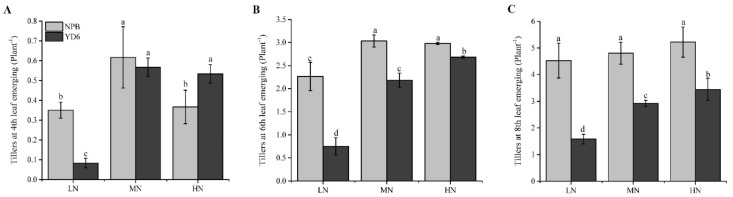
Number of tillers under different N rates in NPB and YD6. (**A**) Tillers at fourth leaf stage; (**B**) Tillers at sixth leaf stage; (**C**) Tillers at eighth leaf stage. The mean and standard deviation (SD) were from three biological replicates, different lowercase letters represent significant differences by Stduent’s *t* test between the treatments at *p* ≤ 0.05.

**Figure 2 ijms-20-05922-f002:**
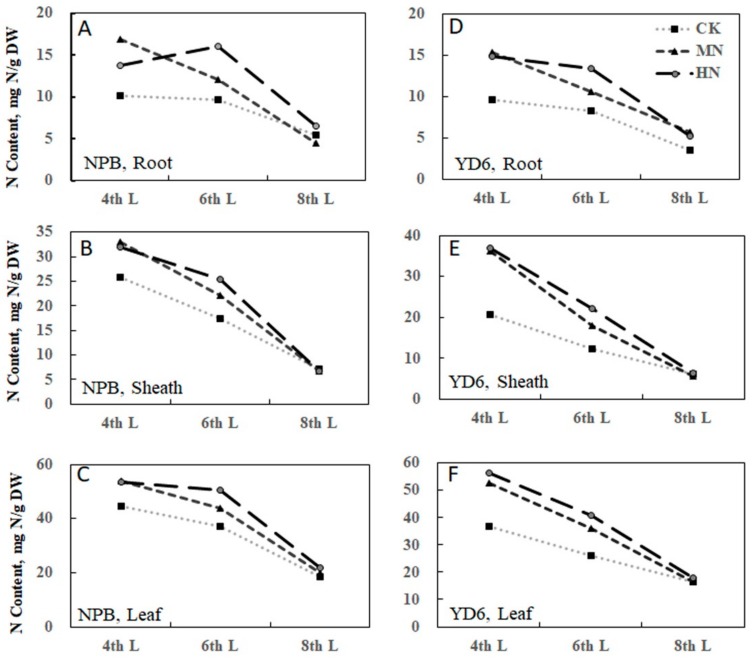
Nitrogen (N) content in root, sheath and leaf organs in different stage of NPB and YD6. N content in root of NPB (**A**) and YD6 (**D**); N content in sheath of NPB (**B**) and YD6 (**E**); N content in leaf of NPB (**C**) and YD6 (**F**).

**Figure 3 ijms-20-05922-f003:**
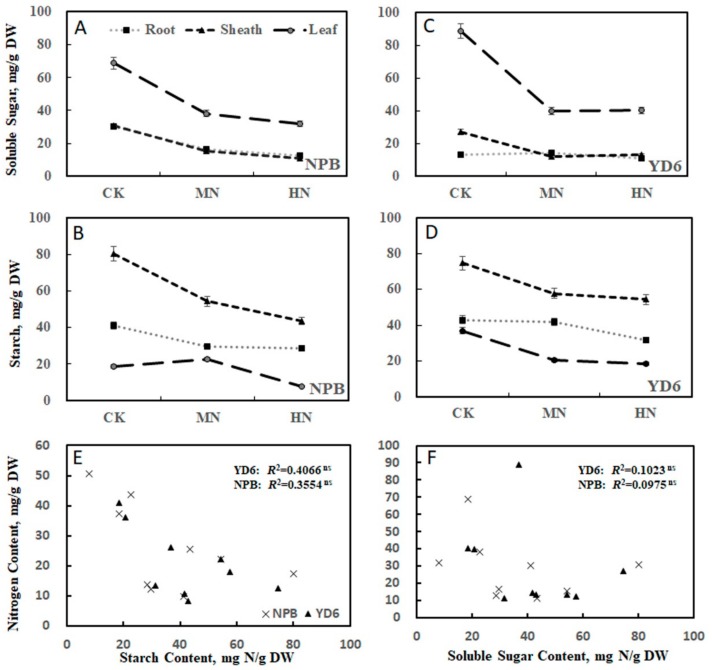
Soluble sugar and starch under different N rates in NPB and YD6. Soluble sugar in NPB (**A**) and YD6 (**C**); Starch in NPB (**B**) and YD6 (**D**); Correlation between N content and starch (**E**), N content and soluble sugar (**F**).

**Figure 4 ijms-20-05922-f004:**
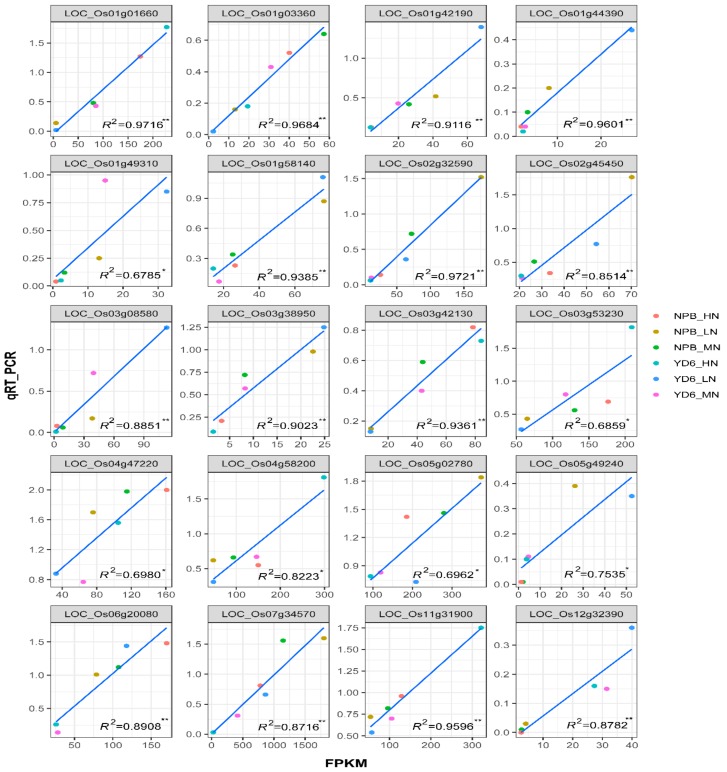
Correlation of qRT-PCR validation and RNA-seq results in 20 representative genes. * and ** represent significance at *p* ≤ 0.05 and *p* ≤ 0.01, respectively.

**Figure 5 ijms-20-05922-f005:**
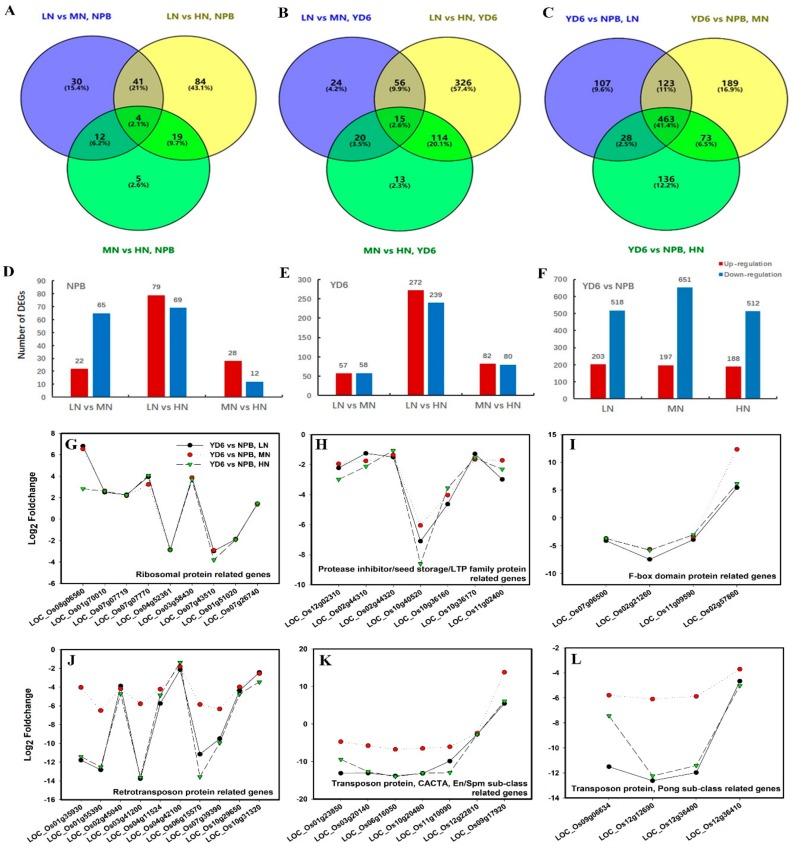
Profiles of DEGs between the comparisons. (**A**,**D**) DEGs in response to N rate in NPB; (**B**,**E**) DEGs in response to N rate in YD6; (**C**,**F**) DEGs between the varieties; (**G–L**) Fold changes of DEGs related to ribosomal protein (**G**), Protease inhibitor/seed storage/LTP family protein (**H**), F-box domain protein (**I**), retrotransposon protein, transposon protein (**J**), CACTA, En/Spm sub-class protein, transposon protein (**K**), Pong sub-class protein (**L**).

**Figure 6 ijms-20-05922-f006:**
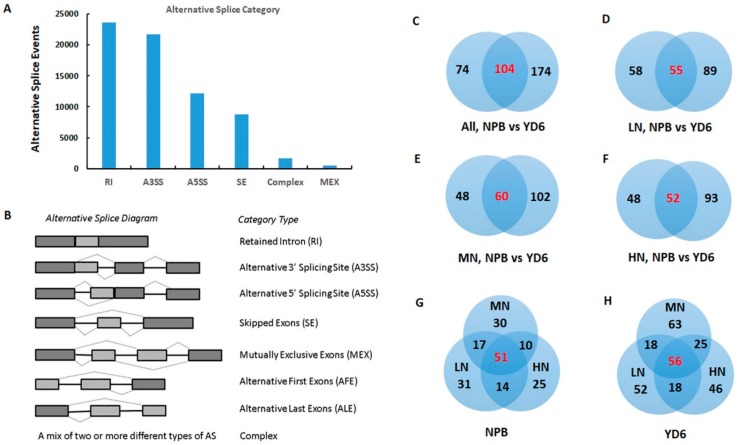
Profiles of alternative splice (AS) and novel genes. (**A,B**) Events and types of AS; (**C–H**) Novel genes between varieties in general (**C**); at LN (**D**); at MN (**E**); at HN (**F**); Novel genes in response to N rate in NPB (**G**); Novel genes in response to N rate in YD6 (**H**).

**Figure 7 ijms-20-05922-f007:**
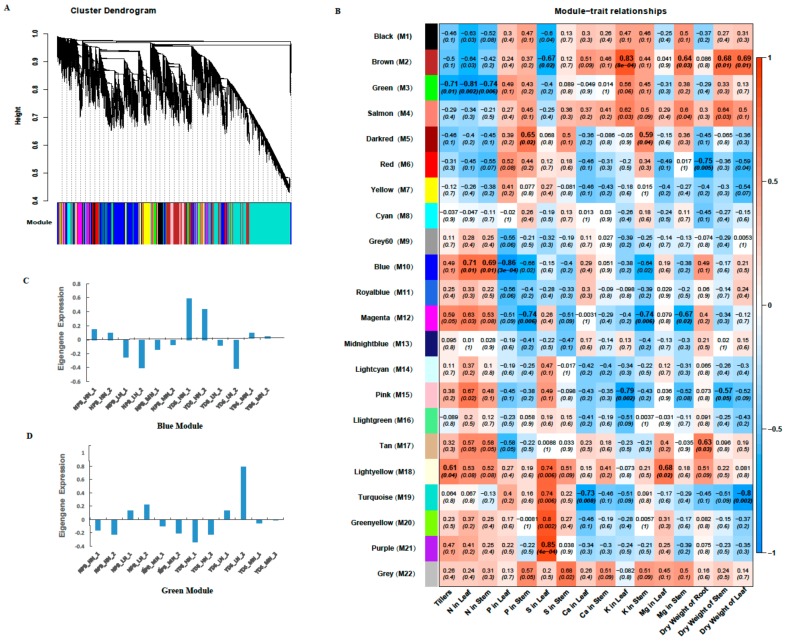
Weighted gene co-expression network analysis (WGCNAs) of gene expressions and the related traits. (**A**) Hierarchical cluster tree showing 22 co-expression modules identified by WGCNAs. Each branch in the tree represents an individual gene; (**B**) Matrix of module – trait correlation: A total of 22 modules shown on the left and the relevance color scale from −1 to 1 is on the right. The numbers in parentheses represent the significance (*p*), and the numbers above represent the correlation coefficient (*r*). The eigengene expression profile in blue module (**C**) and green module (**D**).

**Figure 8 ijms-20-05922-f008:**
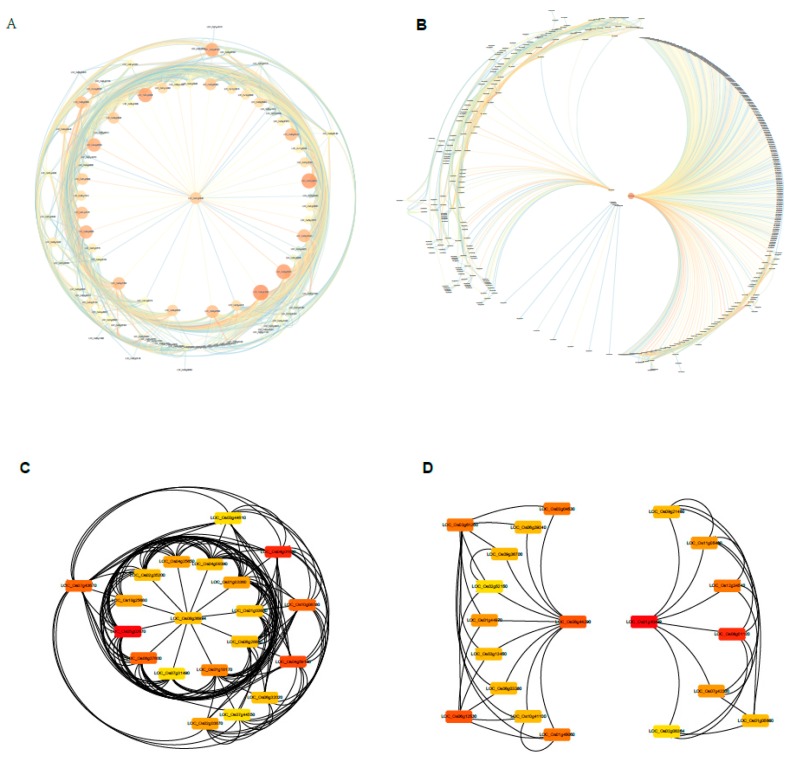
Co-expression network analysis of tiller number related modules. (**A**,**B**) Gene co-expression networks of positively correlated lightyellow module (**A**) and negatively correlated green module (**B**) visualized using Cytoscape software platform. The circle size of and color depth indicate the degree of connectivity; (**C**,**D**) The correlation networks of top 20 nodes in lightyellow module (**C**) and green module (**D**). The color depth represents the number of associated nodes.

**Figure 9 ijms-20-05922-f009:**
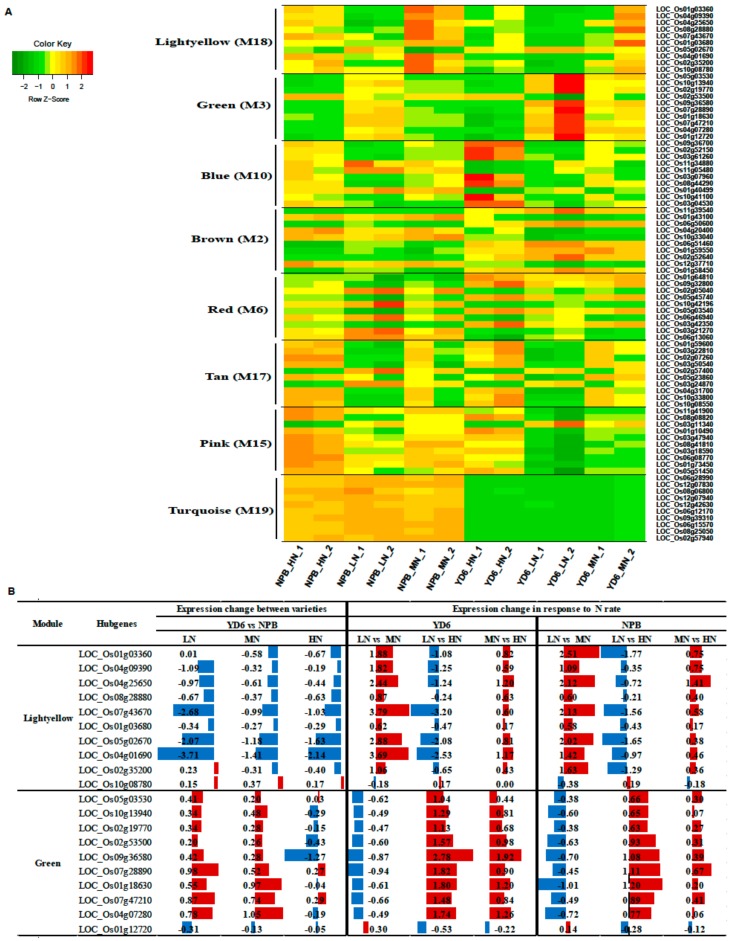
Gene expression of top ten node hub genes of eight related modules. (**A**) Heatmap showing the expression profiles of the top ten node hub genes in each module; (**B**) Fold changes (log2) of the top ten node hub genes in lightyellow and green module. Red and blue color represent scale of up-and down-regulation, respectively.

**Figure 10 ijms-20-05922-f010:**
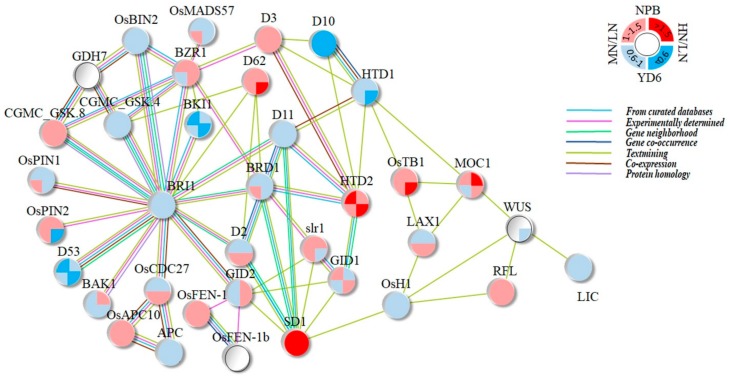
The graphical visualization of protein and protein interaction (PPI) network show the relationship of tiller genes. The gene expression in their fold change are represented by the up and down half of the circle, for NPB and YD6, respectively; MN/LN denotes fold change of MN treatment to LN, HN/LN for HN treatment to LN, represented by the left and right half of the circle, respectively; red and pink mean induction (up regulation), blue and pale blue for suppression, and white for not detected; line colors denote that their connections are generated from curated database (blue), experimentally determined (purple), gene neighborhood (green), textmining (yellow green), co-expression (brown), or protein homology (light purple).

**Table 1 ijms-20-05922-t001:** Candidate hub genes related to the tiller numbers, N rate in leaf and in stem.

Gene Name	Description	K_ME_ Value
**Light yellow module (M18) with positive correlation associated with the tillers**		
LOC_Os04g09390	HEV3 - Hevein family protein precursor, expressed	0.97
LOC_Os01g03680	BBTI8 - Bowman-Birk type bran trypsin inhibitor precursor, expressed	0.95
LOC_Os08g28880	patatin, putative, expressed	0.94
LOC_Os10g08780	expressed protein	0.94
LOC_Os01g03360	BBTI5 - Bowman-Birk type bran trypsin inhibitor precursor, expressed	0.94
LOC_Os02g35200	VP15, putative, expressed	0.88
LOC_Os07g43670	ribonuclease T2 family domain containing protein, expressed	0.85
LOC_Os04g01690	pyridoxal-dependent decarboxylase protein, putative, expressed	0.85
LOC_Os04g25650	cysteine-rich receptor-like protein kinase, putative, expressed	0.85
LOC_Os05g02670	kinesin motor domain containing protein, putative, expressed	−0.80
**Green module (M3) with negative correlation associated with the tillers, N rate in leaf and stem**		
LOC_Os10g13940	MATE efflux protein, putative, expressed	0.98
LOC_Os07g28890	ethylene-responsive protein related, putative, expressed	0.98
LOC_Os01g12720	protein kinase domain containing protein, expressed	0.98
LOC_Os07g47210	GDSL-like lipase/acylhydrolase, putative, expressed	0.97
LOC_Os01g18630	aspartic proteinase oryzasin-1 precursor, putative, expressed	0.95
LOC_Os02g19770	eukaryotic translation initiation factor 1A, putative, expressed	0.94
LOC_Os09g36580	thaumatin, putative, expressed	0.93
LOC_Os05g03530	tetraspanin family protein, putative, expressed	0.90
LOC_Os04g07280	AGAP002737-PA, putative, expressed	0.90
LOC_Os02g53500	RFC5 - Putative clamp loader of PCNA, replication factor C subunit 5, expressed	−0.92
**Blue module (M10) with positive correlation associated with the N rate in leaf and stem**		
LOC_Os08g44290	RNA recognition motif containing protein, putative, expressed	0.98
LOC_Os03g61260	ribosomal L18p/L5e family protein, putative, expressed	0.98
LOC_Os03g04530	cytochrome P450, putative, expressed	0.95
LOC_Os09g36700	ribonuclease T2 family domain containing protein, expressed	0.94
LOC_Os02g52150	heat shock 22 kDa protein, mitochondrial precursor, putative, expressed	0.91
LOC_Os03g07960	expressed protein	0.88
LOC_Os10g41100	CCT motif family protein, expressed	0.84
LOC_Os11g34880	NB-ARC domain containing protein, expressed	−0.70
LOC_Os11g05480	transcription factor, putative, expressed	−0.76
LOC_Os01g40499	S-locus lectin protein kinase family protein, putative, expressed	−0.82

**Table 2 ijms-20-05922-t002:**
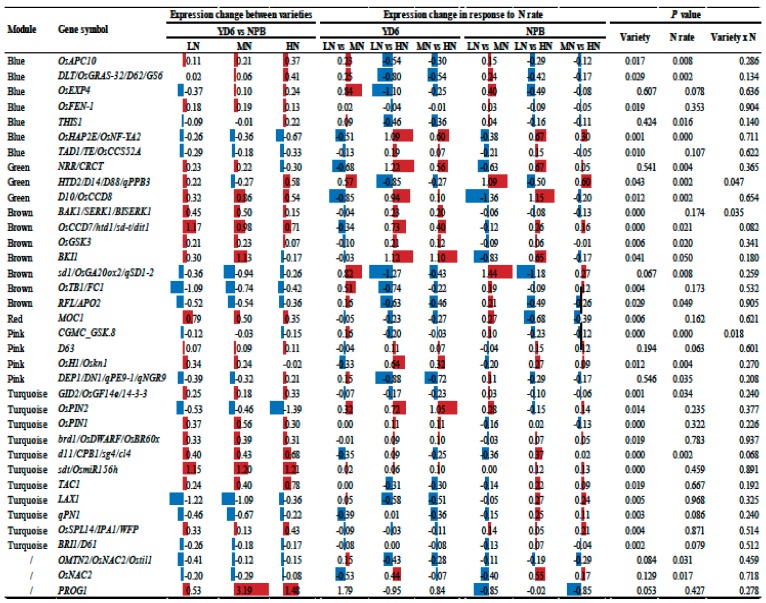
Fold change of the expression level change of tiller genes, their significance, and present module.

Note: The expression changes are in the form of log2, and significances at their respective probability level. Red and blue color represent scale of up-and down-regulation, respectively.

**Table 3 ijms-20-05922-t003:**
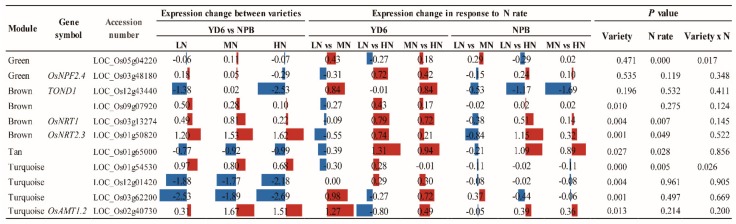
Fold change of the expression level of N metabolism and transporter genes between the varieties and their responses to the N rate.

Note: The expression changes are in the form of log2, and significances at their respective probability level. Red and blue color represent scale of up-and down-regulation, respectively.

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
