# Peer review of "Transcriptomic and Co-Expression Network Profiling of Shoot Apical Meristem Reveal Contrasting Response to Nitrogen Rate between Indica and Japonica Rice Subspecies"

_ijms, 2019, doi:10.3390/ijms20235922_

Round 1

Reviewer 1 Report

The authors compared one indica and one japonica rice (Oryza sativa) varieties for the capacity of the region in the vicinity of the shoot apical meristem to produce tillers under different nitrogen supply. Tiller production generally correlated with the nitrogen content of the plants and showed a negative correlation with soluble carbohydrate content. An RNA-seq was made to estimate gene expression in tissues surrounding the shoot apical meristem. Coexpression clustering revealed modules of genes correlating well with different traits, such as tiller number and nitrogen content. The study concludes that moderate Nnitrogen fertilization is sufficient to promote tillering and nitrogen excess does not contribute to tiller number.

I find the work interesting and timely, however, the manuscript requires further attention before it can be considered for publication.

Comments:

For the GO analysis, the categories listed in the text (lines 223-234) and supplemental figures, are too general. It would be more meaningful to present some of the enriched subcategories, which usually point to more specific processes and functions. The categories “cellular process” and “cell” are not sufficient to provide information on what is common between the genes in the group.

For the coexpression modules, I would strongly suggest a numbering system to go along the colour code. Colorus alone are difficult to follow and memorize, especially for certain readers with colour-blindness. One example is line 336, where even the authors got confused between “lightyellow” and “lightgreen”.

In materials and Methods it is unclear how the correlations between modules and traits were calculated.

Section 2.9: Please, list in the text which modules were selected for the analysis.

Genes are sometimes referred to as “top ten node genes”, sometimes as “top ten node hub genes”. Is there a difference? This is not entirely clear from the description (lines 360-368 and Figure 9).

The authors analyze the splicing pattern in their RNA-seq dataset, however, this is not present in the Discussion. Would be interesting to compare this with other studies (Li et al., 2013 Plant Physiology, Calixto et al., 2018, Plant Cell, Zhu et al., 2018 BMC Genomics).

It would also be interesting to compare the coexpresion analysis with other recent similar studies (He et al., 2016 Frontiers Plant Sci, Goel et al., 2018 Scientific Reports, Brumbarova and Ivanov, 2019, iScience). For example, homologs of none of the here identified top ten genes appear in the list of stress-response genes in latter study, indicating either species-specific differences in gene expression or potential specificity of the genes identified by the authors to nitrogen-mediated developmental responses.

There are some instances where the text needs to be checked for errors:

Line 61: “Tiller provides more SAM and opportunities to achieve more panicles.” (unclear statement)

Line 273: “phenotypicale” should be “phenotypical”

Line 314: “Sulphur” should be “sulfur”

Line 336: “lightgreen” should be “lightyellow”

Supplemental figures have to be combined in a single file and the references included next to the figure. This will make them far more user-friendly.

Author Response

Dear reviewer,

We appreciate your comments and are grateful for your suggestions. Following your advices, we have made revisions accordingly. Point-by-point responses to your comments are listed below.

Original comments:

Point 1: For the GO analysis, the categories listed in the text (lines 223-234) and supplemental figures, are too general. It would be more meaningful to present some of the enriched subcategories, which usually point to more specific processes and functions. The categories “cellular process” and “cell” are not sufficient to provide information on what is common between the genes in the group.

Response 1: Thank you very much for the comment.

We have added the detailed information for the GO analysis. Please see the lines 236-238 in the revised manuscript.

Point 2: For the coexpression modules, I would strongly suggest a numbering system to go along the colour code. Colorus alone are difficult to follow and memorize, especially for certain readers with colour-blindness. One example is line 336, where even the authors got confused between “lightyellow” and “lightgreen”.

Response 2: Thank you very much for the comment.

Following the recommendations, we have added numbering system(M1-M22)along with the colour code in the modules. Please see the Figure 7, 9 and results section in the revised manuscript.

Point 3: In materials and Methods it is unclear how the correlations between modules and traits were calculated.

Response 3: Many thanks for this comment.

We apologize for the miss. We have explained that the algorithm of the correlations between modules and traits in revised manuscript lines 662-663.

Point 4: Section 2.9: Please, list in the text which modules were selected for the analysis.

Response 4: Many thanks for these suggestions.

We have listed the selected modules in the revised manuscript. Please see the lines 329-330.

Point 5: Genes are sometimes referred to as “top ten node genes”, sometimes as “top ten node hub genes”. Is there a difference? This is not entirely clear from the description (lines 360-368 and Figure 9).

Response 5: Thank you very much for your suggestion.

We are sorry for the misleading spelling mistakes. We meant the same, and have corrected in the revised version. Please see lines 381-382.

Point 6: The authors analyze the splicing pattern in their RNA-seq dataset, however, this is not present in the Discussion. Would be interesting to compare this with other studies (Li et al., 2013 Plant Physiology, Calixto et al., 2018, Plant Cell, Zhu et al., 2018 BMC Genomics).

Response 6: Thank you very much for so careful cues to the details.

We have discussed the AS events in the revised manuscript. However, we have not made any detailed analysis of AS in this report. Hopefully, after careful dig, we can make another report on this aspect. Please see the lines 564-569.

Point 7: It would also be interesting to compare the coexpression analysis with other recent similar studies (He et al., 2016 Frontiers Plant Sci, Goel et al., 2018 Scientific Reports, Brumbarova and Ivanov, 2019, iScience). For example, homologs of none of the here identified top ten genes appear in the list of stress-response genes in latter study, indicating either species-specific differences in gene expression or potential specificity of the genes identified by the authors to nitrogen-mediated developmental responses.

Response 7: Thank you very much for your suggestion.

We have included several lines on the co-expression analysis to relate with other recent publications in the revised manuscript. Please see the lines 508-513.

Point 8: There are some instances where the text needs to be checked for errors:

Line 61: “Tiller provides more SAM and opportunities to achieve more panicles.” (unclear statement)

Line 273: “phenotypicale” should be “phenotypical”

Line 314: “Sulphur” should be “sulfur”

Line 336: “lightgreen” should be “lightyellow”

Response 8: Thank you very much for the details.

We have corrected these confusions or errors. The first sentence is rephrased as “Tillers provide more opportunities to the development of more panicles”. Please see line 63, 278, 322 and 345 in the revised manuscript.

Point 9: Supplemental figures have to be combined in a single file and the references included next to the figure. This will make them far more user-friendly.

Response 9: Thank you very much.

Following the recommendations, we have adjusted the supplemental figures. Please see the Supplement file.

Thank you very much. We appreciate very much of the instructive and constructive suggestions.

Sincerely yours,

Xiaoxiang Zhang

Reviewer 2 Report

The obtained results are of interest to further understand the differential transcriptomic responses of rice to N dose application. The current paper has been well organized and written that should be published. However, authors are requested to make some revisions as follows:

The introduction part should be provided more information involved in N rate in both japonica and indica cultivars,  for instance: https://doi.org/10.1016/j.fcr.2019.107625 and other recently published papers. Materials and Methods should be improved by adding more information in detail (see comments directly highlighted in pdf file) Results: Some figures are poor quality, they should be changed by high quality and high solution Figs to replace them (e.g: Fig 8..). Discussion: explain why did authors select NPB and YB6 varieties in this study, and more discussion on the significant results obtaining in this study. English editing should be judiciously improved, see some edited in the file. References should be carefully formatted following to IJMS style.

Author Response

Dear reviewer,

We appreciate your comments and are grateful for your suggestions. Following your advices, we have made revisions accordingly. Point-by-point responses to your comments are listed below.

Original comments:

Point: The obtained results are of interest to further understand the differential transcriptomic responses of rice to N dose application. The current paper has been well organized and written that should be published. However, authors are requested to make some revisions as follows:

The introduction part should be provided more information involved in N rate in both japonica and indica cultivars, for instance: https://doi.org/10.1016/j.fcr.2019.107625 and other recently published papers. Materials and Methods should be improved by adding more information in detail (see comments directly highlighted in pdf file) Results: Some figures are poor quality, they should be changed by high quality and high solution Figs to replace them (e.g: Fig 8..). Discussion: explain why did authors select NPB and YB6 varieties in this study, and more discussion on the significant results obtaining in this study. English editing should be judiciously improved, see some edited in the file. References should be carefully formatted following to IJMS style.

Response: Thank you very much for the comment.

We have added the information related to N rate in both japonica and indica cultivars in the Introduction section and in M&M section as well. Please see the lines 57-58 and Lines 589-590

In Materials and Methods section, further explanation on the varieties added in the lines 578-580.

In Results section, a PDF version with high quality and high solution of Figure 8 is provided.

In Discussion section, we have included a few more words on why we chose to use NPB and YB6, and the significance of these varieties. Please see the lines 470-472.

We have polished a bit of the languages in revised manuscript. The style of references has been formatted following to IJMS style. We apologize for the mess in our first manuscript due to our ignorance. Please see the revised manuscript.

Thank you very much. We appreciate very much of the instructive and constructive suggestions.

Sincerely yours,

Xiaoxiang Zhang